# Defective nucleotide-dependent assembly and membrane fusion in Mfn2 CMT2A variants improved by Bax

Nyssa B Samanas, Emily A Engelhart, Suzanne Hoppins

**Mitofusins are members of the dynamin-related protein family of large GTPases that harness the energy from nucleotide hydrolysis to remodel membranes. Mitofusins possess four structural domains, including a GTPase domain, two extended helical bundles (HB1 and HB2), and a transmembrane region. We have characterized four Charcot-Marie-Tooth type 2A–associated variants with amino acid substitutions in Mfn2 that are proximal to the hinge that connects HB1 and HB2. A functional defect was not apparent in cells as the mitochondrial morphology of Mfn2-null cells was restored by expression of any of these variants. However, a significant fusion deficiency was observed in vitro, which was improved by the addition of crude cytosol extract or soluble Bax. All four variants had reduced nucleotide-dependent assembly in cis, but not trans, and this was also improved by the addition of Bax. Together, our data demonstrate an important role for this region in Mfn2 GTP-dependent oligomerization and membrane fusion and is consistent with a model where cytosolic factors such as Bax are masking molecular defects associated with Mfn2 disease variants in cells.**

## Introduction

Mitochondrial dynamics have become increasingly recognized as an important indicator of and contributor to both cellular health and death. Mitochondrial shape and their cellular distribution change during the cell cycle, in response to stress, and as part of apoptosis (Tondera et al, 2009; Scorrano, 2013; Labbé et al, 2014; Horbay & Bilyy, 2016). Mitochondria are trafficked on microtubules and the overall shape and connectivity of the mitochondrial network is maintained or modified through mitochondrial fusion and division, which are mediated by membrane remodeling large GTPase proteins of the dynamin related protein (DRP) family (Labbé et al, 2014). At steady state, it is estimated that mitochondrial fusion and division events are balanced. When mitochondrial division events exceed fusion events, the network fragments into many small individual mitochondria, and this fragmentation is associated with mitophagy and apoptosis. In contrast, when mitochondrial fusion occurs more frequently than division, the result is a more connected network comprising longer mitochondria, which is associated with increased ATP production, such as during a cellular stress response. The importance of these processes and their regulation is highlighted by the association of dysregulated mitochondrial dynamics with various diseases such as Parkinson's disease, diabetes, and peripheral neuropathies (Züchner et al, 2004; Vital & Vital, 2012; Celardo et al, 2014; Wada & Nakatsuka, 2016; Rovira-Llopis et al, 2017).

Mitochondrial DRP-mediated fusion is poorly understood and is mechanistically distinct from both SNARE- and viral-mediated fusion. Fusion DRPs that reside in the outer and inner mitochondrial membranes are mitofusin 1, and mitofusin 2 (Mfn1 and Mfn2) and Opa1, respectively. Mfn1 and Mfn2 are functionally related but nonredundant paralogs in mammalian cells (Santel & Fuller, 2001; Chen et al, 2003; Eura, 2003; Ishihara et al, 2004). Interestingly, mutations in *MFN2*, but not *MFN1*, are the main cause of the peripheral neuropathy Charcot-Marie-Tooth syndrome type 2A (CMT2A) (Züchner et al, 2004). In CMT2A patients, distal nerve degeneration leads to weakness, sensory loss, gait impairment, and foot deformations (Gemignani & Marbini, 2001).

The mitochondrial fusion defects associated with some CMT2A-associated variants of Mfn2 can be functionally complemented by Mfn1 (Detmer & Chan, 2007; Engelhart & Hoppins, 2019). Consistent with this, a recent report found that expression of exogenous Mfn1 in neurons of a CMT2A mouse model rescued axonal degradation (Zhou et al, 2019). The importance of the interaction between the mitofusin paralogs is further highlighted by the observation that the optimal fusion complex is composed of both Mfn1 and Mfn2 (Hoppins et al, 2011). As with other DRPs, both mitofusins exhibit nucleotide-dependent self-assembly, and the ability to form higher order oligomers has been correlated with fusion activity (Engelhart & Hoppins, 2019). Further supporting a role for assembly in DRP-mediated fusion, intermolecular complementation has been observed between nonfunctional variants of the yeast mitofusin homolog Fzo1 that possess amino acid substitutions in distinct functional domains (Griffin & Chan, 2006). Consistently, Fzo1 has been shown to form a large docking ring that generates an extensive area of contact at the interface of two mitochondria (Brandt et al, 2016).

Department of Biochemistry, University of Washington, Seattle, WA, USA

Correspondence: shoppins@uw.edu

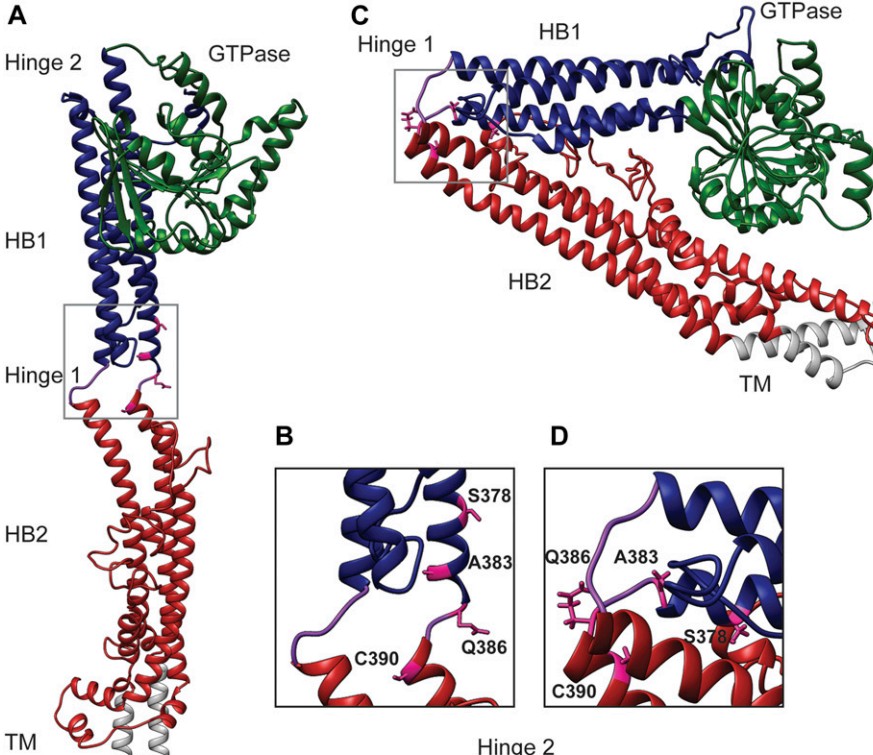

Mitofusins contain four major structural domains, including the GTPase domain, two sequential extended helical bundles (HB1 and HB2) connected by flexible loops, and a transmembrane domain (Fig 1). The domain organization is similar to bacterial dynamin like protein (BDLP), which has been structurally characterized (Low & Löwe, 2006; Low et al, 2009). In BLDP, the relative positions of these domains changes in different nucleotide states via rearrangements in the hinge region that connect the domains, resulting in either an extended or a closed state (Fig 1A and B). It is hypothesized that mitofusin proteins undergo similar structural rearrangements around hinge regions (Jimah & Hinshaw, 2019). Indeed, atomic structures of a minimal GTPase domain construct of Mfn1 demonstrate that relative to the GTPase domain, HB1 can exist in two distinct, nucleotide-dependent conformations due to changes in hinge 2 (Qi et al, 2016; Cao et al, 2017; Yan et al, 2018). Furthermore, mini-peptides or small molecules thought to alter the stability of the extended or closed state of mitofusin change the overall structure of the mitochondrial network in cells (Franco et al, 2016; Rocha et al, 2018).

The function of the mitofusin helical bundles is relatively unexplored. To identify functionally informative variants of mitofusin and characterize the contribution of this region, we exploited the fact that amino acid substitutions in Mfn2 are associated with human disease. Several of these CMT2A-associated substitutions found in heterozygous patients occur near hinge 1, consistent with the prediction that this domain is functionally important. To gain insight into the role of this region in mitofusin-mediated membrane fusion, we interrogated the mitochondrial fusion activity and biochemical properties of CMT2A-associated hinge 1-proximal variants. Our data indicate that the integrity of this region is required for optimal mitochondrial fusion activity and efficient nucleotide-dependent assembly of Mfn2.

# Results

## Mfn2 hinge 1-proximal variants restore reticular mitochondrial morphology in Mfn2-null cells

We set out to characterize four mutant variants of Mfn2 with amino acid substitutions proximal to hinge 1: S378P, A383V, Q386P, and C390F (Fig 1 and Table S1). The precise position of these residues is not known. In a structural model of Mfn2 based on the closed structure of BDLP (PDB 2J69), S378 resides in HB1, C390 is within HB2, and A383 and Q386 are within a loop connecting HB1 and HB2 (Fig 1A). BDLP also exists in an extended conformation in the presence of lipid and GMPPNP (PDB 2W6D) and using this model, A383 is predicted to be in a helix of HB1 with S378 (Fig 1B). In a recently published structure of truncated human Mfn2, all four positions are located in α helix 3 of HB1, adjacent to the truncation site where the N terminus stops and is linked to a C-terminal helix (Fig S1) (Li et al, 2019). Therefore, these positions are likely to be in a conformationally dynamic region proximal to or within hinge 1. We began by analyzing mitochondrial morphology in stable cell lines expressing the CMT2A versions of Mfn2. To create these lines, we introduced either wild-type or mutant *MFN2* with a C-terminal 3×FLAG tag into Mfn2-null MEFs using retroviral transduction (Chen et al, 2003).

**Figure 1. Structural model of the positions of four hinge 1-proximal amino acid substitutions associated with CMT2A.**
**(A)** Structural model of the predicted extended structure of Mfn2 based on the crystal structure of the structurally related protein BDLP with 5′-Guanylyl imidodiphosphate (GMPPNP) (protein data base [PDB] 2W6D). The GTPase domain is green, HB1 is blue, HB2 is red, the transmembrane (TM) domain is grey, and Loops 1/2 are purple. Structural prediction performed by I-TASSER server (Zhang, 2009; Yang & Zhang, 2015). **(B)** Enlarged view of hinge 1 showing the positions of relevant CMT2A-related amino acids in pink with side chains. **(C)** Structural model of the predicted closed structure of Mfn2 based on the crystal structure of the structurally related protein BDLP with GDP (PDB 2J69). **(A)** Domains are colored as described in (A). Structural prediction performed by I-TASSER server (Zhang, 2009; Yang & Zhang, 2015). **(D)** Enlarged view of Loop 1 from hinge 1 showing the positions of the CMT2A-related amino acids in pink with side chains.

Clonal populations expressing Mfn2 at similar levels were selected for characterization (Fig S2).

In wild-type MEFs, mitochondria were in reticular networks where most of the mitochondria were longer than 2.5 $\mu$m (Fig 2, Mfn1$^{+/+}$Mfn2$^{+/+}$). Mfn2-null cells transduced with an empty vector, conversely, contained mainly fragmented individual mitochondria less than 2.5 $\mu$m in length (Fig 2, vector), which is consistent with published observations of Mfn2-null cells (Chen et al, 2003). Expression of wild-type Mfn2 in Mfn2-null cells restored fusion activity and a reticular mitochondrial network in about 80% of cells (Fig 2, Mfn2$^{WT}$). Somewhat surprisingly, all Mfn2 mutant variants examined here (Mfn2$^{S378P}$, Mfn2$^{A383V}$, Mfn2$^{Q386P}$, and Mfn2$^{C390F}$) also restored a reticular mitochondrial network in Mfn2-null cells to a similar extent as Mfn2$^{WT}$. These results indicate that all four of these Mfn2 variants are stable and well-folded and support robust fusion activity in Mfn2-null fibroblasts. This is consistent with previously reported data that showed subsets of CMT2A-associated mutations restored fusion activity in Mfn2-null cells (Detmer & Chan, 2007; Engelhart & Hoppins, 2019).

Mitochondrial connectivity was further assessed using a mitochondrial matrix–targeted photoactivatable GFP (mito-PAGFP). After activation of mito-PAGFP within a 1-$\mu$m square, the localization of mito-PAGFP relative to the rest of the mitochondrial network was monitored to assess mitochondrial connectivity, movement, and fusion. In wild-type cells, we observed that the proportion of the mitochondrial network with activated mito-PAGFP increased ~2.5 fold 50 min after activation following photoactivation (Fig S3). The same analysis was performed in Mfn2-null MEFs expressing an Mfn2 CMT2A variant and although we observe a slightly smaller fold increase than the wild-type, the difference was not statistically significant (Fig S3).

Because we observed only small changes in mitochondrial connectivity under baseline conditions, we also challenged these cells with oxidative stress, which has previously been shown to result in a hyperfused mitochondrial network in about a third of the population (Shutt et al, 2012). After treatment of cells with diamide, we scored mitochondrial morphology in wild-type cells and Mfn2-null MEFs expressing these Mfn2 CMT2A variants. We observed a similar degree of mitochondrial hyperfusion in all cell populations, also consistent with our conclusion that mitochondrial fusion is not significantly impaired in cells (Fig S4).

## Mfn2 hinge 1-proximal variants have an in vitro mitochondrial fusion defect

We reasoned that these substitutions may result in biochemical changes to the protein that were masked in the context of the cell,

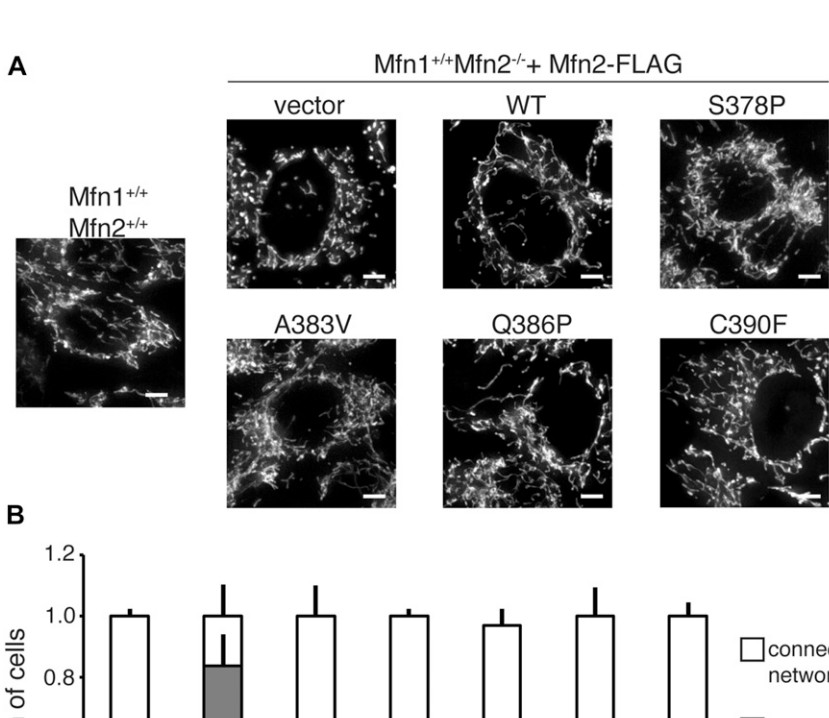

**Figure 2. Mfn2 hinge 1-proximal variants support mitochondrial fusion when expressed in Mfn2-null cells.**
**(A)** Representative images of mitochondrial networks in wild-type (Mfn1$^{+/+}$Mfn2$^{+/+}$) or Mfn2-null (Mfn1$^{+/+}$Mfn2$^{-/-}$) MEFs expressing the indicated Mfn2 variant. Mitochondria were stained with MitoTracker Red CMXRos and visualized by fluorescence microscopy. Images represent a maximum intensity projection. Scale bars = 5 $\mu$m. **(A, B)** Quantification of mitochondrial morphology in cells represented in (A). Error bars indicate mean + SD from three independent, blinded experiments (n ≧ 100 cells per population per experiment).

where many factors modulate the structure of the mitochondrial network. Therefore, we proceeded to characterize these mutant forms of Mfn2 in the context of isolated mitochondria. To quantify the mitochondrial fusion activity of the Mfn2 variants in the absence of cytosolic factors, we used a cell-free mitochondrial fusion assay. Mitochondria isolated from cells expressing red fluorescent protein (RFP) or CFP targeted to the mitochondrial matrix were mixed, incubated in fusion buffer and then imaged by fluorescence microscopy. Fusion events were scored as the overlap of the two fluorophores in three dimensions. All assays were performed in parallel with mitochondria isolated from wild-type cells, and data are expressed as a percent of wild-type controls. Mitochondria isolated from Mfn2-null cells transduced with empty vector fused at a much lower frequency than wild-type controls (Fig 3A). Fusion of mitochondria isolated from the clonal population of Mfn2-null cells expressing Mfn2$^{WT}$ was similar to wild-type controls (Fig 3A), consistent with the restoration of the mitochondrial morphology in cells. To quantify the fusion activity of the Mfn2 hinge 1-proximal variants in vitro, mitochondria were isolated from each of the clonal populations described above. In each case, the mitochondrial fusion activity was significantly lower than wild-type controls (Fig 3A).

The in vitro mitochondrial fusion data revealed that amino acid substitutions in this region diminish the fusion activity of Mfn2. Given that we could not detect a robust mitochondrial fusion defect in cells, we predicted that a cytosolic factor could be enhancing the fusion activity of the mutant variants. To test this in vitro, we performed the mitochondrial fusion assay in the presence of cytosol from wild-type MEFs. As has been previously reported, the addition of the cytosol-enriched fraction to wild-type mitochondria moderately stimulated fusion activity (Hoppins et al, 2011; Shutt et al, 2012). In contrast, the cytosol-enriched fraction did not alter the fusion activity of mitochondria isolated from Mfn2-null cells, indicating that Mfn2, but not Mfn1, is regulated by the cytosolic factor (Fig 3B). The addition of cytosol also increased the fusion activity of mitochondria that possess the Mfn2 hinge 1-proximal variants similarly to wild-type controls (Fig 3B). These data indicate that the fusion defect associated with the hinge variants can be improved by cytosolic factors in cells.

We have previously demonstrated that soluble Bax stimulates mitochondrial fusion in vitro and in cells (Hoppins et al, 2011). To determine if Bax can improve the fusion activity of the CMT2A-associated variants, we performed the in vitro fusion assay in the presence of purified Bax. As expected, 300 nM Bax was sufficient to stimulate in vitro fusion of wild-type mitochondria (Fig 3C, Mfn1$^{+/+}$ Mfn2$^{+/+}$). In addition, Bax stimulated mitochondrial fusion by all of the Mfn2 variants. Therefore, we predict that in cells, Bax is facilitating mitochondrial fusion by these CMT2A-associated variants of Mfn2.

### Mfn2 hinge 1-proximal mutant variants interact with Mfn1 in cis and in trans

Mfn1 and Mfn2 physically interact in the same membrane, in cis, and across two membranes, in trans, as measured by co-immunoprecipitation (Chen et al, 2003; Scott Detmer & Chan, 2007; Engelhart & Hoppins, 2019). To determine if these Mfn2 CMT2A variants interact with Mfn1 in cis or in trans, we tested whether Mfn1

would co-immunoprecipitate with Mfn2-FLAG. To distinguish between cis and trans, we mixed mitochondria that possess endogenous Mfn1 and Mfn2-FLAG with mitochondria that possess Mfn1-eGFP and endogenous Mfn2 (Fig 4A). In this way, three unique interactions with Mfn2-FLAG could be assessed, as diagramed in Fig 4A. First, we can detect an interaction between Mfn2-FLAG and endogenous Mfn1 in cis (Fig 4); second, Mfn2-FLAG can interact with Mfn1-eGFP in trans; third, endogenous Mfn2 could interact with Mfn2-FLAG in trans. These immunoprecipitation reactions were performed in the presence of the GTP transition state mimic GDP-BeF$_3$, which has been shown to stabilize an interaction between mitofusin molecules and to promote a tethering interaction (Qi et al, 2016; Yan et al, 2018; Engelhart & Hoppins, 2019). As a negative control, the same immunoprecipitation reactions were performed in the absence of GDP (BeF$_3$ only). As expected, wild-type Mfn2-FLAG immunoprecipitated both endogenous Mfn1 and Mfn1-eGFP (Fig 4B, black arrow and open arrowhead, respectively). The amount of endogenous Mfn1 interacting in cis was not significantly different in the presence or absence of the transition state mimic. In contrast, the trans interaction with Mfn1-eGFP is more robust in the presence of GDP-BeF$_3$ compared with BeF$_3$ alone (Fig 4B and C). Together, these data indicate that only the trans interaction between Mfn1 and Mfn2 is highly dependent on the nucleotide-binding state of the mitofusin proteins. Each of the four Mfn2 hinge 1-proximal variants also immunoprecipitated Mfn1 in cis and in trans (Fig 4B and C), indicating that there is no defect in the physical interaction with Mfn1 in either context. Importantly, we observe no Mfn1-eGFP in the absence of Mfn2-FLAG (Fig S5). Interestingly, even in the presence of GDP-BeF$_3$, we could not detect an interaction in trans between Mfn2-FLAG and endogenous Mfn2 (Fig 4B, white arrow), which suggests that full-length Mfn2 does not form a robust homotypic trans complex.

### Nucleotide-dependent oligomerization is diminished in Mfn2 hinge 1-proximal variants

Defects in mitofusin assembly correlate with reduced rates of mitochondrial fusion in vitro (Engelhart & Hoppins, 2019). To determine the capacity of these Mfn2 variants to form oligomers, we used blue native gel electrophoresis (BN-PAGE). Mitochondria were left untreated or were incubated with either GTP or the non-hydrolyzable analog GMPPNP. Mitochondria were then lysed and separated by BN-PAGE and subject to Western blot analysis. When mitochondria were untreated, most Mfn2$^{WT}$ migrated in oligomers that approximately correspond in size to a dimer (Fig 5A and C, arrow), with some protein migrating as larger assemblies (Fig 5A and C, arrowheads). When these mitochondria were instead incubated with nucleotide, the ratio of dimer to larger assemblies decreased, consistent with nucleotide-dependent assembly of Mfn2 dimers into higher order oligomers. Specifically, after incubation with GTP, more Mfn2$^{WT}$ migrated in two larger assemblies, with a notable increase in the ~450-kD oligomer (Fig 5A, open arrowhead). Incubation of mitochondria with GMPPNP also promoted higher order assembly with enhanced stability of a ~320-kD oligomer and some protein migrating as the ~450-kD oligomer (Fig 5A, closed and open arrowheads, respectively). Together, these data indicate that Mfn2 exists primarily as a dimer in the mitochondrial

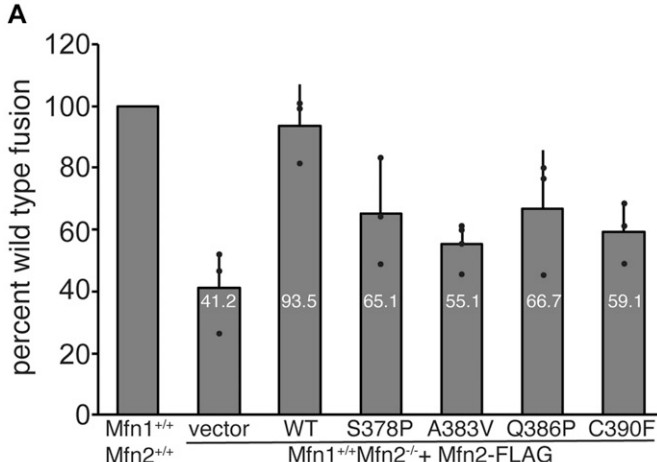

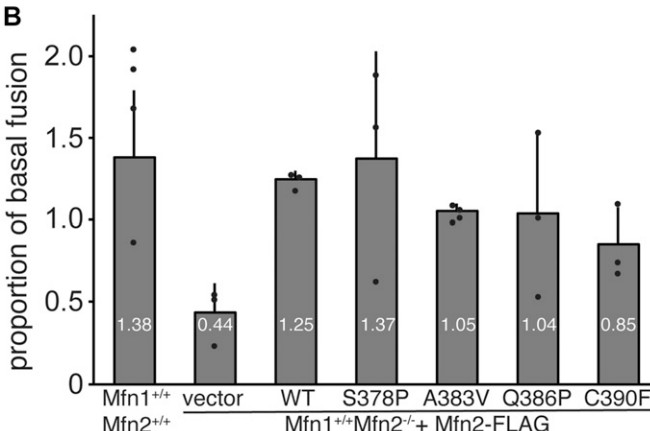

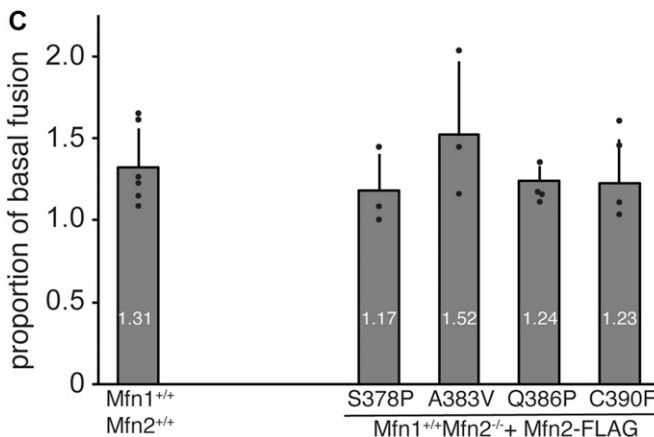

**Figure 3. Mitochondrial in vitro fusion assay reveals a defect for all Mfn2 hinge 1-proximal variants.**
**(A)** Mitochondria isolated from wild-type cells or clonal populations of Mfn2-null MEFs either transduced with empty vector or expressing the indicated Mfn2 variant were subject to in vitro fusion conditions at 37°C for 60 min. The data are represented as relative to wild-type controls performed in parallel. Error bars indicate mean + SD from at least three independent experiments. The average values of each is indicated in white within the individual bars. **(B)** Mitochondrial in vitro fusion assay performed as in (A) except with the addition of cytosol-enriched fraction to the reaction buffer. Data are represented as proportion of control reactions performed in parallel without cytosol. Error bars indicate mean + SD from at least three independent experiments. The average values of each is indicated in white within the individual bars. **(C)** Mitochondrial in vitro

outer membrane and assembles into at least two larger oligomeric species in a nucleotide-dependent manner.

In untreated mitochondria, all of these Mfn2 variants migrated similarly to wild type, with most of the protein in the dimer (Fig 5A). In the presence of GTP, the variants also assembled similar to wild type, with the exception of Mfn2$^{S378P}$, which had less protein migrating as the ~450-kD oligomer compared with wild type. Each of the hinge-proximal variants displayed altered assembly relative to wild-type Mfn2 in the presence of GMPPNP (Fig 5B). Compared with Mfn2$^{WT}$, Mfn2$^{S378P}$, Mfn2$^{A383V}$, Mfn2$^{Q386P}$, and Mfn2$^{C390F}$, all showed a significantly decreased amount of protein migrating as the ~320-kD oligomer (Fig 5B). These data indicate that amino acid substitutions in this region prevent the stable assembly of Mfn2 into this oligomeric species in the presence of the non-hydrolyzable GTP analog.

Given that these mutant variants also exhibit an in vitro mitochondrial fusion defect, we hypothesize that nucleotide-dependent assembly contributes to efficient Mfn2-mediated membrane fusion. Our data would further predict that Bax, which improved in vitro fusion activity, could facilitate Mfn2 assembly. To test this, we assessed Mfn2 nucleotide-dependent assembly in the presence of recombinant purified Bax, which has been previously demonstrated to promote Mfn2-dependent mitochondrial fusion and Mfn2 assembly (Karbowski et al, 2006; Hoppins et al, 2011). In the presence of Bax, we observe more Mfn2$^{WT}$ in the ~320-kD oligomer, consistent with Bax playing a role in Mfn2 assembly (Fig 5C and D). Furthermore, Mfn2$^{S378P}$, Mfn2$^{A383V}$, Mfn2$^{Q386P}$, and Mfn2$^{C390F}$ also had increased abundance of the ~320-kD oligomer in the presence of Bax (Fig 5C and D). Together, these data support the conclusion that impaired nucleotide-dependent assembly results in diminished in vitro fusion activity and that cytosolic factors such as Bax compensate for these defects in cells.

## Double mutations reveal that HB1 and HB2 contribute independently to mitochondrial fusion

Our data indicate that this region of Mfn2 is important for Mfn2 fusion activity. Based on the BDLP structural models, S378 is predicted to be located in HB1, whereas C390 is part of HB2. We considered the possibility that these two distinct structural domains function cooperatively to control the conformational state of Mfn2. If this is the case, a variant of Mfn2 with amino acid substitutions in both HB1 and HB2 would be predicted to have a more severe functional defect than variants with two substitutions in the same structural domain, as has been observed for Fzo1 (De Vecchis et al, 2017). To test this hypothesis, we used Mfn2$^{L710P}$, a CMT2A-associated variant predicted to be in HB2, near Loop 2 in Hinge 1 (Verhoeven et al, 2006) (Figs 1 and S6). This amino acid substitution has been characterized in Mfn1, Mfn1$^{L691P}$, which had some fusion activity in Mfn1-null cells (Koshiba et al, 2004). We generated double-mutant variants of Mfn2 with this mutation and either

---

fusion assay performed as in (A) except with the addition of 300 nM purified Bax protein to the reaction buffer. Data are represented as proportion of control reactions performed in parallel without Bax. Error bars indicate mean + SD from at least three independent experiments. The average values of each is indicated in white within the individual bars.

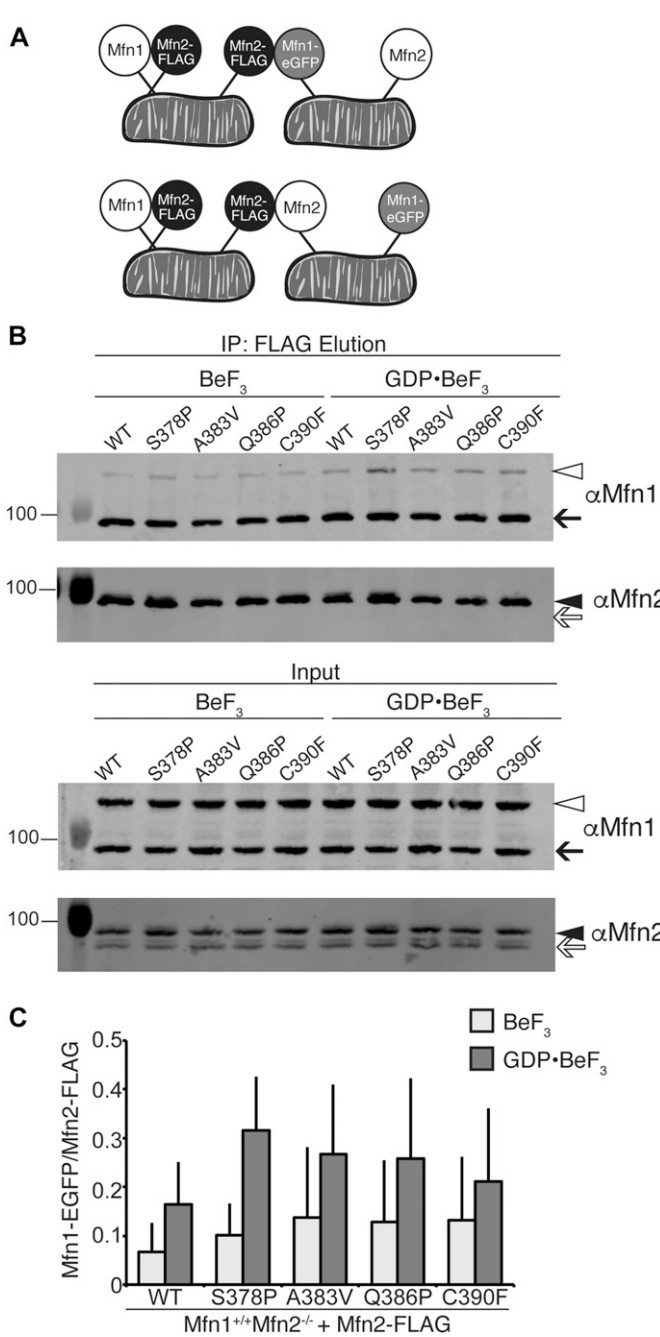

**Figure 4. Mfn2 hinge 1-proximal variants interact with Mfn1 in cis and trans.**
**(A)** Schematic of the differential epitope labeling used in the co-immunoprecipitation assay. Interactions tested are Mfn2-FLAG with Mfn1 in cis (top left), Mfn2-FLAG with Mfn1-eGFP in trans (top center), and Mfn2-FLAG with Mfn2 in trans (bottom center). **(B)** Mitochondria were isolated from a clonal population of Mfn1-null cells expressing Mfn1$^{WT}$-eGFP at endogenous levels and clonal populations of Mfn2-null MEFs expressing the indicated Mfn2-FLAG variant. Mitochondria that possess Mfn1-eGFP and Mfn2 were combined with mitochondria that possess Mfn1 and Mfn2-FLAG; these mixtures were incubated with BeF$_3$ in the absence or presence of GDP. After lysis, immunoprecipitation was performed with $\alpha$-FLAG magnetic beads. Proteins eluted from the beads were subjected to SDS–PAGE and immunoblotting with $\alpha$-Mfn1 and $\alpha$-Mfn2, as indicated. Arrows indicate endogenous Mfn1 (black) and Mfn2 (white); arrowheads indicate Mfn1-eGFP (white) and Mfn2-FLAG (black). Input represents 3% of the input and elution represents 37.5% of the immunoprecipitated protein. **(C)** Quantification of the percentage of Mfn1-eGFP in the elution compared with Mfn2-FLAG is shown as the mean + SD of three independent experiments.

Mfn2$^{S378P}$, which is located in HB1 or Mfn2$^{C390F}$, which is located in HB2 (Mfn2$^{S378P/L710P}$ and Mfn2$^{C390F/L710P}$, respectively).

To determine the fusion activity of these double-mutant variants, we used retroviral transduction to express Mfn2-mNeonGreen (Mfn2-NG) in Mfn2-null MEFs, which has previously been shown to restore fusion activity (Engelhart & Hoppins, 2019). The mitochondrial morphology was scored in cells expressing Mfn2, as assessed by colocalization of Mfn2-NG with MitoTracker Red. As expected, Mfn2$^{WT}$, Mfn2$^{S378P}$, and Mfn2$^{C390F}$ restored the reticular mitochondrial network, which recapitulated our observations from the clonal populations described above (Fig 6, compare with Fig 2). Expression of Mfn2$^{L710P}$ also restored a reticular network in ~65% of cells (Fig 6), indicating a mild defect in mitochondrial fusion, consistent with previous reports (Koshiba et al, 2004).

In Mfn2-null cells expressing the HB2 double-mutant Mfn2$^{C390F/L710P}$, the mitochondrial morphology was comparable with that in cells expressing Mfn2$^{L710P}$ alone (Fig 6). In contrast, when both HB1 and HB2 possess an amino acid substitution, there is significantly less mitochondrial fusion. Specifically, in Mfn2-null cells expressing Mfn2$^{S378P/L710P}$, most cells possessed a mitochondrial network that was either fragmented or in fragmented aggregates, and only ~30% of cells had a reticular mitochondrial network. Together, these results support a model where HB1 and HB2 provide unique and separate contributions to support Mfn2-mediated membrane fusion.

## Discussion

In this study, we performed an in-depth characterization of CMT2A-associated amino acid substitutions located proximal to or within Hinge 1 of Mfn2, which connects HB1 and HB2. In other DRPs, hinges mediate conformational changes that are required for membrane remodeling activity. The results presented here are consistent with BDLP-based structural models that suggest this region functions as a hinge. The recently published structure of truncated human Mfn2 raises the intriguing possibility that these four amino acids could also be positioned in a continuous helix ($\alpha$ helix 3) adjacent to the predicted hinge (Li et al, 2019). The integrity of this helix could also be important to support conformational changes in the context of the full-length protein. In either case, our data implicate this region in GTP-dependent oligomerization of Mfn2 and highlight that these CMT2A-associated variants possess unique oligomerization characteristics. These molecular defects also correspond to a decrease in mitochondrial fusion efficiency in vitro. Both defects were improved by the addition of cytosolic factors, specifically purified Bax, which indicates that mitofusin function is subject to complex regulation in cells.

Our data indicate that Mfn2 exists primarily as a dimer in the mitochondrial outer membrane, and this form is not altered by the amino acid substitutions tested here. Furthermore, the Mfn2 variants are able to interact with Mfn1 on the same and adjacent mitochondria in a nucleotide-dependent manner similarly to wild-type Mfn2. Together, these data suggest that these amino acid substitutions do not alter protein folding or stability of Mfn2. We demonstrate that GTP stabilizes a ~450-kD oligomer, whereas a non-hydrolyzable analog stabilizes a ~320-kD oligomer. In the presence of GTP, most of the hinge variants are similar to wild type as each can form both dimers

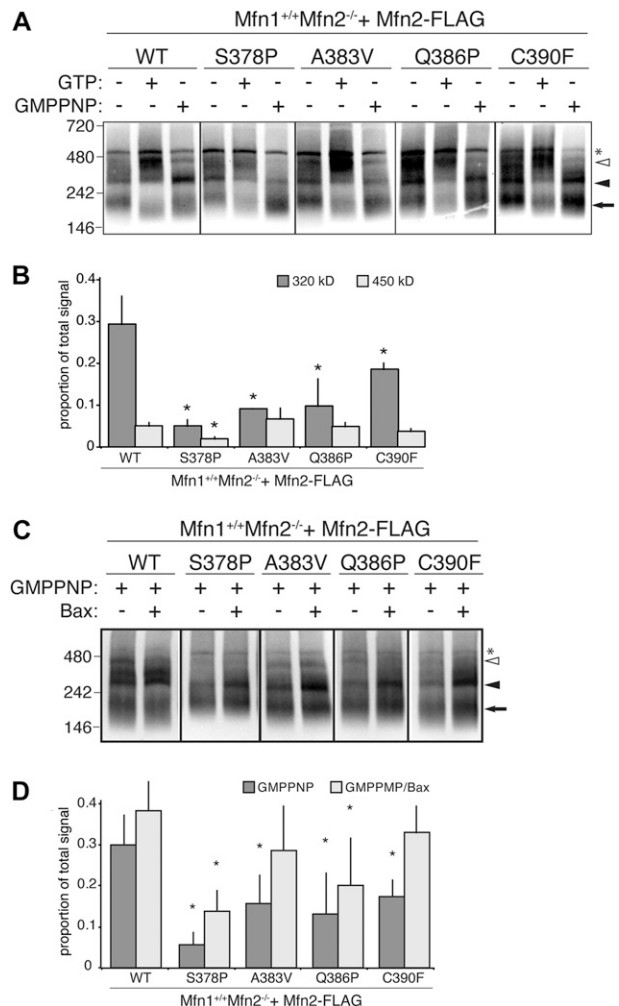

**Figure 5. Mfn2 hinge 1-proximal variants have altered nucleotide-dependent assembly.**
**(A)** Mitochondria were isolated from clonal populations of Mfn2-null MEFs expressing the indicated Mfn2 variant. Mitochondria were either untreated or incubated with 2 mM GTP or 2 mM GMPPNP as indicated before lysis and separation by BN-PAGE followed by immunoblotting with anti-FLAG antibody. Arrow indicates predicted dimer, closed arrowhead indicates ~320-kD species, and open arrowhead indicates ~450-kD species. Asterisk (*) indicates nonspecific signal. Approximate molecular weights are in kD on the left. **(B)** Quantification of proportion of the total protein observed in the ~320 or ~450-kD band after treatment with GMPPNP (filled arrowhead). Error bars indicate mean + SD from at least three independent experiments and the statistical significance were determined by paired $t$ test analysis between the indicated data and wild type (*$P < 0.05$). **(C)** Mitochondria were prepared as in (A) and incubated with 2 mM GMPPNP with or without 1 μM recombinant purified Bax. Arrow indicates predicted dimer, closed arrowhead indicates ~320-kD species, and open arrowhead indicates ~450-kD species. Asterisk (*) indicates nonspecific signal. Approximate molecular weights are in kD on the left. **(D)** Quantification of proportion of the total protein observed in the ~320-kD band. Error bars indicate mean + SD from at least three independent experiments and the statistical significance were determined by paired $t$ test analysis between the indicated data and wild type (*$P < 0.05$).

and ~450-kD oligomers. This indicates that these variants are not unable to fold or oligomerize. In contrast, the hinge-proximal variants do not readily form the ~320-kD state adopted when Mfn2 is bound to GMPPNP. Given that all of these Mfn2 variants co-immunoprecipitate

Mfn1 as efficiently as wild-type Mfn2, interaction with Mfn1 is not likely to play a role in this assembly. Indeed, previously published data from our laboratory suggest that the BN-PAGE assemblies are homo-oligomers (Engelhart & Hoppins, 2019). These data are consistent with the hypothesis that the nucleotide-binding state of Mfn2 alters the conformational state of the protein, which results in the preferential formation of distinct oligomeric species. Therefore, the reduced assembly observed for these variants could be due to aberrant structural rearrangements of HB1 and HB2 as mediated by hinge 1. Our data further implicate Bax in the formation or stability of these oligomers.

Functional complementation has been observed between two nonfunctional alleles of Fzo1, the mitofusin homolog (Griffin & Chan, 2006). These data indicate that different molecules in the fusion complex can contribute different properties. In the full-length protein structural models based on BDLP, the amino acid substitutions tested here are predicted to be within either HB1 (S378), HB2 (C390), or a flexible loop between the two (A383V and Q386P). Given that each individual amino acid substitution was associated with a similar partial loss of function, we considered that these structural domains may work cooperatively to evoke nucleotide-dependent conformational changes. To test this, we assessed the fusion activity of variants with two amino acid sub-stitutions, either in the same domain (HB2 and C390F & L710P) or in different domains (HB1, S378P & HB2, and L710P). When amino acid substitutions were both in HB2, the fusion activity in cells was more substantial than when the substitutions were in HB1 and HB2. Therefore, our data indicate that distinct structural domains provide unique contributions to Mfn2 fusion activity within the same molecule. As a whole, the data presented here are consistent with the hypothesis that different nucleotide-dependent conformations support either the formation or stabilization of different assembly states and that the integrity of the predicted hinge region that connects HB1 and HB2 plays a role in this process. We have further shown that this assembly is supported not only by intramolecular interactions but also by regulating cytosolic factors such as Bax.

# Materials and Methods

### Cell culture

All cells were grown at 37°C and 5% $CO_2$ and cultured in DMEM (Thermo Fisher Scientific) containing 1× GlutaMAX (Thermo Fisher Scientific) with 10% FBS (Seradigm) and 1% penicillin/streptomycin (Thermo Fisher Scientific). MEF cells (Mfn wild-type and Mfn2-null) were purchased from American Type Culture Collection.

### Retroviral transduction and generation of clonal populations

Plat-E cells (Cell Biolabs) were maintained in complete media supplemented with 1 μg/ml puromycin and 10 μg/ml blasticidin and plated at ~80% confluency the day before transfection. Plat-E cells were transfected with FuGENE HD (Promega) and transfection regent was incubated overnight before a media change. Viral supernatants were collected at ~48, 56, 72, and 80 h post-

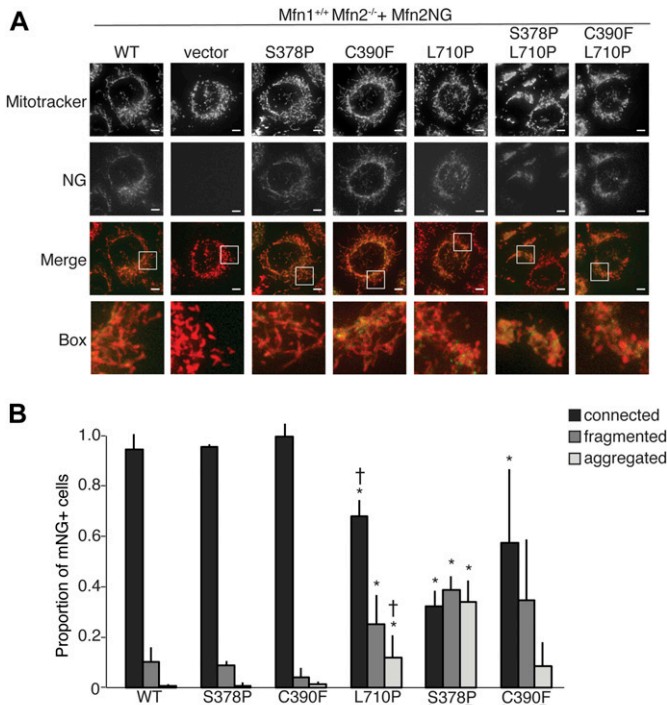

**Figure 6. Mfn2 variants with substitutions in both HB1 and HB2 are defective for fusion in Mfn2-null cells.**
**(A)** Representative images of mitochondrial networks in Mfn2-null MEFs expressing the indicated Mfn2-mNeonGreen variant. Mitochondria were stained with MitoTracker Red CMXRos and visualized by fluorescence microscopy. Images represent a maximum intensity projection. Scale bars = 5 μm. **(B)** Quantification of mitochondrial morphology of cells represented in (A). Error bars indicate mean + SD from at least three independent experiments and the statistical significance was determined by paired $t$ test analysis between the indicated data and wild type (*$P <$ 0.05) or between the indicated data and Mfn2[S378P/L710P] (†$P <$ 0.05).

transfection and incubated with MEFs in the presence of 8 mg/ml polybrene. ~16 h after the last viral transduction, MEF cells were split and selection medium was added if needed (1 μg/ml puromycin or 200 μg/ml hygromycin).

Clonal populations were generated by plating cells at very low density and clones were collected onto sterile filter paper dots soaked in trypsin. After expansion, whole-cell extracts from clonal populations were screened by Western blot analysis for mitofusin against wild-type controls.

### Transfection and microscopy

All cells were plated in No. 1.5 glass-bottomed dishes (MatTek). MEFs were incubated with 0.1 μg/ml MitoTracker Red CMXRos (Invitrogen) for 15 min at 37°C with 5% $CO_2$, washed, and incubated with complete media for at least 45 min before imaging. For analysis of mitochondrial morphology under oxidative stress conditions, 100 μM diamide was added to the culture media for 1 h before imaging. MEFs were imaged at 37°C with 5% $CO_2$. A Z-series with a step size of 0.3 μm was collected with a Nikon Ti-E widefield microscope with a 63× NA 1.4 oil objective (Nikon), a solid-state light source (Spectra X; Lumencor), and an sCMOS camera (Zyla 5.5 Megapixel). Each cell

line was imaged by a blinded researcher on at least three separate occasions (n > 100 cells per experiment).

### Photo-activatable mitochondrial (mt)-GFP

Cells transduced with mito-PAGFP (#23348; Addgene) were plated in No. 1.5 glass-bottomed dishes (MatTek). MEFs were incubated with 0.1 μg/ml MitoTracker Red CMXRos (Invitrogen) for 15 min at 37°C with 5% $CO_2$, washed, and incubated with complete media for at least 45 min before imaging. MEFs were imaged at 37°C with 5% $CO_2$. A region that was ~1 μm² was activated using a 405-nm laser, and the same cell was imaged after 50 min. Images were collected with a Nikon Ti-E widefield microscope with a 63× NA 1.4 oil objective (Nikon), a solid-state light source (Spectra X; Lumencor), and an sCMOS camera (Zyla 5.5 Megapixel).

### Image analysis

Images were deconvolved using 8–15 iterations of 3D Landweber deconvolution. Deconvolved images were then analyzed using Nikon Elements software. Maximum intensity projections were created using Photoshop (Adobe). Mitochondrial morphology was scored as follows: reticular indicates that fewer than 30% of the mitochondria in the cell were fragments (fragments defined as mitochondria less than 2 μm in length); fragmented indicates that most of the mitochondria in the cell were less than 2 μm in length; aggregated indicates fragmented mitochondria that were not distributed throughout the cytosol.

### Preparation of mitochondria or cytosol-enriched fraction

For each experiment, three to five 15-cm plates each of MEFs were grown to ~90% confluency. The cells were harvested by cell scrapping, pelleted, and washed in mitochondrial isolation buffer (MIB) (0.2 M sucrose, 10 mM Tris-MOPS [pH 7.4], and 1 mM EGTA). The cell pellet was resuspended in one cell pellet volume of cold MIB, and cells were homogenized by 10–14 strokes on ice with a Kontes Potter-Elvehjem tissue grinder set at 400 rpm. The homogenate was centrifuged (500$g$, 5 min, 4°C) to remove nuclei and unbroken cells, and homogenization of the pellet fraction was repeated, followed by centrifugation for 5 min at 500$g$ and 4°C. The supernatant fractions were combined and centrifuged again for 5 min at 500$g$ and 4°C to remove remaining debris. The supernatant was transferred to a clean microfuge tube and centrifuged (7,400$g$, 10 min, 4°C) to pellet a crude mitochondrial fraction. The post-mitochondrial supernatant fraction was saved as the cytosol-enriched fraction. The crude mitochondrial pellet was resuspended in a small volume of MIB. Protein concentration of fractions was determined by Bradford assay (Bio-Rad Laboratories).

### In vitro mitochondrial fusion

An equivalent mass (12.5 μg) of mtTagRFP and mtCFP mitochondria were mixed, washed in 500 μl MIB, and concentrated by centrifugation (7,400$g$, 10 min, 4°C). After a 10-min incubation on ice, the supernatant was removed, and the mitochondrial pellet was resuspended in 10 μl fusion buffer (20 mM PIPES-KOH [pH 6.8], 150 mM KOAc, 5 mM Mg(OAc)$_2$, 0.4 M sorbitol, 0.12 mg/ml creatine

phosphokinase, 40 mM creatine phosphate, 1.5 mM ATP, and 1.5 mM GTP) or 10 $\mu$l cytosol-enriched buffer (2.5 $\mu$l of the cytosol-enriched fraction obtained from WT MEFs and 7.5 $\mu$l fusion buffer). Fusion reactions were incubated at 37°C for 60 min.

## Analysis of mitochondrial fusion

Mitochondria were imaged on depression microscope slides by pipetting 4 $\mu$l fusion reaction onto a 3% low-melt agarose bed, made in modified fusion buffer (20 mM PIPES-KOH [pH 6.8], 150 mM KOAc, 5 mM Mg(OAc)$_2$, and 0.4 M sorbitol). A Z-series of six 0.2-$\mu$m steps was collected with a Nikon Ti-E widefield microscope with a 100× NA 1.4 oil objective (Nikon), a solid-state light source (Spectra X; Lumencor), and an sCMOS camera (Zyla 5.5 Megapixel). For each condition tested, mitochondrial fusion was assessed by counting ≥300 total mitochondria per condition from ≥4 images per condition (50–200 mitochondria per image collected), and fusion was scored by colocalization of the red and cyan fluorophores in three dimensions.

## BN-PAGE

Isolated mitochondria (15–30 $\mu$g) were incubated with or without 2 mM nucleotide (GTP or GMPPNP) with or without 1 $\mu$M purified recombinant Bax as indicated in modified MIB (0.2 M sucrose, 10 mM Tris-MOPS [pH 7.4], 1 mM EGTA, 5 mM Mg(OAc)$_2$, 50 mM KOAc, 1× HALT protease inhibitor [Thermo Fisher Scientific], 0.5 mM PMSF) at 37°C for 30 min. Lysis buffer (50 mM Bis-Tris, 50 mM NaCl, 10% wt/vol glycerol, and 0.001% Ponceau S, pH 7.2) was added to this mixture so that the final concentration of digitonin was 1% weight/vol, and this was incubated for 15 min on ice. Lysates were centrifuged at 16,000$g$ at 4°C for 30 min. The cleared lysate was mixed with Invitrogen NativePAGE 5% G-250 Sample Additive to a final concentration of 0.25%. The samples were separated on a Novex NativePAGE 4–16% Bis-Tris Protein Gels (Invitrogen) at 4°C. The gels were run at 40 V for 30 min and then at 100 V for 30 min with dark cathode buffer (1× NativePAGE Running Buffer [Invitrogen], 0.02% [wt/vol] Coomassie G-250). Dark cathode buffer was replaced with light cathode buffer (1× NativePAGE Running Buffer [Invitrogen] and 0.002% [wt/vol] Coomassie G-250), and the gel was run at 100 V for 30 min and subsequently at 250 V for 60–75 min until the dye front ran off the gel. After electrophoresis was complete, the gels were transferred to the Polyvinylidene fluoride (PVDF) membrane (Bio-Rad Laboratories) at 30 V for 16 h in transfer buffer (25 mM Tris, 192 mM glycine, and 20% methanol). The membranes were incubated with 8% acetic acid for 15 min and washed with H$_2$O for 5 min. The membranes were dried at 37°C for 20 min and then rehydrated in 100% methanol and washed in H$_2$O. Membranes were blocked in TBST + 4% milk for 20 min and were probed with anti-FLAG (Sigma-Aldrich) for 4 h at room temperature or overnight at 4°C in TBST + 4% milk. The membranes were incubated with HRP-linked secondary antibody (Cell Signaling Technology) at room temperature for 1 h. They were developed in SuperSignal Femto ECL reagent (Thermo Fisher Scientific) for 5 min and imaged on iBright Imaging System (Thermo Fisher Scientific). Band intensities were quantified using ImageJ software (NIH). NativeMark Unstained Protein Standard (Life Technologies) was used to estimate molecular weights of mitofusin protein complexes.

## Co-immunoprecipitation

Differentially tagged isolated mitochondrial populations (50 $\mu$g each) were mixed together. Mitochondria were incubated at 37°C for 30 min with beryllium fluoride (2.5 mM BeSO$_4$ and 25 mM NaF) with or without 2 mM GDP in fusion buffer (20 mM PIPES-KOH [pH 6.8], 150 mM KOAc, 5 mM Mg(OAc)$_2$, 0.4 M sorbitol with 0.12 mg/ml creatine kinase, 40 mM creatine phosphate, and 1.5 mM ATP). Mitochondria were solubilized in lysis buffer (20 mM HEPES-KOH [pH 7.4], 50 mM KCl, and 5 mM MgCl$_2$) with 1.5% wt/vol n-dodecyl $\beta$-D-maltoside (DDM), and 1× Halt Protease Inhibitor (Thermo Fisher Scientific) for 30 min on ice. The lysates were cleared at 10,000$g$ for 15 min at 4°C. The supernatant was incubated with 50 ml magnetic $\mu$MACS Anti-DYKDDDDK MicroBeads (Miltenyi Biotec) for 30 min on ice. The sample was applied to a MACS Column (Miltenyi Biotec) placed in the magnetic field using a $\mu$MACS Separator (Miltenyi Biotec); washed once with 300 $\mu$l 20 mM HEPES-KOH [pH 7.4], 50 mM KCl, 5 mM MgCl$_2$, and 0.1% DDM; and once with 200 $\mu$l 20 mM HEPES-KOH [pH 7.4], 50 mM KCl, and 5 mM MgCl$_2$. One column volume (25 $\mu$l) SDS–PAGE loading buffer (60 mM Tris–HCl [pH 6.8], 2.5% sodium dodecyl sulfate, 5% $\beta$ME, 5% sucrose, and 0.1% bromophenol blue) was incubated for 15 min at room temperature and proteins were eluted once with 40 $\mu$l SDS–PAGE loading buffer. Samples were run on an SDS–PAGE gel and transferred onto nitrocellulose membranes at 94V for 1 h in 1× transfer buffer (25 mM Tris, 192 mM glycine, and 20% methanol). The membranes were blocked in TBST + 4% milk for at least 45 min and were probed with anti-Mfn1 antibody and anti-Mfn2 antibody for 4 h at room temperature or overnight at 4°C. The membranes were incubated with DyLight secondary antibody (Invitrogen) at room temperature for 1 h. The membranes were imaged on LI-COR Imaging System (LI-COR Biosciences).

## Western blot analysis of clonal populations

Protein lysates from MEFs were obtained by resuspending PBS-washed cells in radioimmunoprecipitation assay (RIPA) lysis buffer (150 mM NaCl, 1% Nonidet P-40, 1% sodium deoxycholate, 0.1% SDS, 25 mM Tris [pH 7.4], and 1× Halt Protease Inhibitor Cocktail, EDTA-free [Thermo Fisher Scientific]). The samples were incubated on ice for 5 min and then spun at 21,000$g$ for 15 min at 4°C. The supernatant was transferred to a clean tube, and protein concentration was measured by bicinchoninic acid assay (Thermo Fisher Scientific). The samples were run on an SDS–PAGE gel and transferred onto nitrocellulose membranes at 100 V for 50 min in 1× transfer buffer. The membranes were blocked in TBST + 4% milk for at least 45 min and were probed with anti-Mfn1, anti-Mfn2 (Sigma-Aldrich), anti-VDAC (Invitrogen), or anti-$\alpha$ Tubulin (Invitrogen) antibody for 4 h at room temperature or overnight at 4°C. The membranes were incubated with DyLight secondary antibody (Invitrogen) at room temperature for 1 h. The membranes were imaged on LI-COR Imaging System (LI-COR Biosciences).

## Bax expression and purification

Bax was purified as previously described (Suzuki et al, 2000). Briefly, pTYB1-Bax was expressed in *Escherichia coli* strain BL21(DE3) grown in Luria–Bertani medium with 150 $\mu$g/ml ampicillin at 37°C to an

OD600 ~ 0.6, and protein expression was induced by the addition of 1 mM isopropyl 1-thio-β-D-galactopyranoside. Induced cultures were grown for about 3 h at 37°C and then the cells were harvested and frozen in liquid nitrogen and stored at −80°C. The pellet was resuspended in TEN buffer (20 mM Tris–HCl [pH 8.0], 1 mM EDTA, and 500 mM NaCl), and the cells were lysed using a microfluidizer (Avestin). The lysate was subjected to centrifugation at 14,000 rpm for 45 min, and the cleared lysate was passed through a 0.2-$\mu$m filter before binding to a chitin column equilibrated with TEN buffer. The column was washed with TEN buffer and then incubated with TEN buffer + 30 mM DTT for 48 h at 4°C. Cleaved protein was eluted with TEN buffer, and buffer exchange was performed with 20 mM Tris–HCl (pH 8.0) with 10% glycerol. Protein was bound to Q-Sepharose and eluted with a linear gradient of NaCl. tris(2-carboxyethyl)phosphine (TCEP) was added to Bax-containing fractions to a final concentration of 1 mM before the protein was aliquoted and frozen at −80°C.

### Plasmids & primers

The following plasmids were purchased from Addgene: pBABE-hygro (#1765), pBABE-puro (#1764), mito-PAGFP (#23348), pclbw-mito TagRFP (#58425), pclbw-mitoCFP (#58426). The following primers were used to for site-directed mutagenesis by Gibson Assembly:

Mfn2$^{S378P}$ F: (5′-CCGTTCGTCTCATCATGGATCCCCTGCACATCGCAGC-3′)
Mfn2$^{S378P}$ R: (5′-GCTGCGATGTGCAGGGGATCCATGATGAGACGAACGG-3′)
Mfn2$^{A383V}$ F: (5′-ATTCCCTGCACATCGCAGTTCAGGAGCAGCGGG-3′)
Mfn2$^{A383V}$ R: (5′-CCCGCTGCTCCTGAACTGCGATGTGCAGGGAAT-3′)
Mfn2$^{Q386P}$ F: (5′-GCACATCGCAGCTCAGGAGCCGCGGGTTTATTGCCTA-GAAATGCGG-3′)
Mfn2$^{Q386P}$ R: (5′-CCGCATTTCTAGGCAATAAACCCGCGGCTCCTGAGCTGC-GATGTGC-3′)
Mfn2$^{C390F}$ F: (5′-GGGTTTATTTCCTAGAAATGCGG-3′)
Mfn2$^{C390F}$ R: (5′-CCGCATTTCTAGGAAATAAACCC-3′)
Mfn2$^{L710P}$ F: (5′-GACATCACCCGAGATAATCCGGAGCAGGAAATTGCTGC-3′)
Mfn2$^{L710P}$ R: (5′-GCAGCAATTTCCTGCTCCGGATTATCTCGGGTGATGTC-3′).

# Supplementary Information

# Acknowledgements

We thank the members of the Hoppins lab and Laura Lackner for critical scientific discussions and critical reading of the manuscript. We would also like to thank Richard Youle for sharing the pTYB1-Bax plasmid and Bax purification protocol. NB Samanas was supported by the National Institute of General Medical Sciences (NIH NIGMS) Training grant No T32GM007270 and S Hoppins is supported by NIH NIGMS Grant No R01GM-118509.

## Author Contributions

NB Samanas: conceptualization, data curation, formal analysis, validation, investigation, visualization, and methodology.

EA Engelhart: investigation and methodology.
S Hoppins: conceptualization, formal analysis, investigation, and methodology.

## Conflict of Interest Statement

The authors declare that they have no conflict of interest.

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
