## [Reviewer comments · Life Science Alliance]

Life Science Alliance

Defective nucleotide dependent assembly and membrane fusion in Mfn2 CMT2A variants improved by Bax

Nyssa Samanas, Emily Engelhart, and Suzanne Hoppins

DOI: <https://doi.org/10.26508/lsa.201900527>

Corresponding author(s): Suzanne Hoppins, University of Washington

Review Timeline:	Submission Date:	2019-08-19
	Editorial Decision:	2019-09-30
	Revision Received:	2020-03-02
	Editorial Decision:	2020-03-24
	Revision Received:	2020-03-24
	Accepted:	2020-03-25

Scientific Editor: Andrea Leibfried

Transaction Report:

September 30, 2019

Re: Life Science Alliance manuscript #LSA-2019-00527

Dr. Suzanne Hoppins
University of Washington
1959 NE Pacific St Health Sciences Building J383
Seattle, WA 98195

Dear Dr. Hoppins,

Thank you for submitting your manuscript entitled "Mfn2 requires Hinge 1 integrity for efficient nucleotide-dependent assembly and membrane fusion" to Life Science Alliance. The manuscript was assessed by expert reviewers, whose comments are appended to this letter.

As you will see, the reviewers are generally supportive of your work and we would thus like to invite you to submit a revised version to us, addressing the points raised by the reviewers. The controls mentioned (rev#2) should get included. Reviewer #1 and #2 point out that a bit more insight into the differences observed in vitro vs in cellulo is needed to provide a significant value to others, and they provide constructive input on how to offer such insight. The approaches proposed by rev#1 and #2 differ however, so I think it would be really good in this case to discuss further in order to define the revision upfront. Maybe you could provide a preliminary point-by-point response to the criticisms raised so that we can define a set of experiments that would address both rev#1 and #2 request for further insight without necessarily following all experimental suggestions made?

Thank you for this interesting contribution to Life Science Alliance. We are looking forward to receiving your revised manuscript.

Sincerely,

B. MANUSCRIPT ORGANIZATION AND FORMATTING:

Reviewer #1 (Comments to the Authors (Required)):

The authors have studied the role of mutations in the Hinge 1 region of Mfn2 involved in disease on the fusion activity of the protein. They report that these mutations decrease the fusion activity and

the oligomerization of the protein in in vitro assays of mitochondria fusion, but mitochondrial shape is not affected in cells. They argue that cytosolic factors contribute to compensate for the fusion defects of the mutants in cells, and indeed show that added Bax is able to increase oligomerization, although this remains as a rather enigmatic observation. Furthermore, the authors show that combination of mutations in both helical bundles of Mfn2 exhibit more severely impaired function. The manuscript is well and clearly written, although the abstract could improve in clarity. The results are in principle of interest for the field, but there are a number of issues that should be addressed before publication.

The authors could explain in more detail which mutations are associated with which diseases

Could the cytosolic factor that masks the lower activity of the Mfn mutants Bax or Drp1? The authors could add the recombinant proteins in the in vitro fusion experiments.

Or could it be that the mitochondrial shape is maintained by changes in dynamics lost in isolated mitochondria? To test this later hypothesis the authors could measure mitochondrial dynamics by FRAP or similar.

The authors state: "Together, these data support the conclusion that impaired nucleotide-dependent assembly results in diminished in vitro fusion activity and that cytosolic factors such as Bax compensate for these defects in cells." However they only show correlation, not causality. And for Bax, only a compensation in assembly, not fusion activity, which could easily be tested as mentioned above.

It is surprising that the authors do not find any mitochondrial defect in the cells expressing the Mfn mutants, given that they are associated with disease. One possibility could be that this is due to the cellular system used, which does not recapitulate the situation in neurons. To test this hypothesis, the authors could check the activity of the Mfn mutants in neurons or neuron-like cells. In addition, they should check if additional mitochondrial functions other than fusion are affected in cells expressing the Mfn mutants, such as metabolism, calcium signalling, mitophagy or apoptosis.

Reviewer #2 (Comments to the Authors (Required)):

In the present manuscript, the authors have analyzed the role of CMT2A disease mutants of Mfn2, located at the hinge 1 region, based on a model to the bacterial homologue BDLP. Consistent with previous results from the Dorn and Cohen/Taly labs, among others, they suggest that dynamic conformational changes and oligomerization properties in Mfn2 constitute a critical feature for their functionality. They newly show that cellular compensatory mechanisms allow maintaining mitochondrial reticulation of hinge 1 mutants, despite defects observed when using an in vitro fusion assay. This is concomitant with alterations in mitofusin oligomerization of all hinge 1 mutants analyzed. Thus, the title provided is a totally fair judgment of their results, which are equally globally convincing. However, for a strong conceptual advance, it would be important to provide a mechanistic justification for the "opposing" phenotypes of the CMT2A variants in vivo and in vitro, and resulting disease-implications.

Specific points to be addressed:

Do the four hinge 1 mutants (S378P, A383V, Q386P and C390F) alter fusion in vivo? This should be measured, e.g. with PA-GFP assays.

In Fig. 3, which is the justification for clear alterations in fusion, as demonstrated by the assay in vitro? How do stronger Mfn1-Mfn2 interactions in trans (Fig. 4) (perhaps reflecting longer resident times before hydrolysis) relate to the oligomers observed by BN-page in Fig. 5? In other words, what do these oligomers represent? Do the different oligomerization properties of the mutants reflect cis or trans interaction defects? And homo Mfn2 or hetero Mfn1/2 oligomers? This could be addressed by testing the differently tagged variants used for Fig. 4. Consequently, which complex does Bax affect? Cis or trans? Before or after hydrolysis? Measurements of GTP binding and hydrolysis rates could help explaining these mutants.

Further, in Fig. 4, the effect observed by Bax addition for oligomerization is rather mild. A convincing role of Bax in these Mfn2 mutants should be demonstrated in cells, perhaps with experiments reminiscent of their own Mol Cell paper. E.g., how does soluble/mito-inserted/apoptosis-incompetent Bax affect the mitochondrial morphology of these 4 Mfn2 mutants?

In Fig. S1, please include a shorter exposure of both blots. In Fig. 3, panels A and B could be presented with the same scaling, and the average values indicated. In Fig. 4, a Mfn1 and Mfn2 antibody specificity should be included. Plus, a negative control with cells w/o Mfn2-Flag should be performed, to subtract background binding of Mfn1-eGFP to the beads. In Fig. 5A, a negative control w/o Mfn2-Flag should be added. Also, a short exposure of the blots should be included.

To further broaden the conceptual advance, the following questions might be considered: Do their mutants alter fusion/fission balance? Is fission inhibited? Do the mutant cells respond/interact differently to Drp1, Drp1 receptors, or Fis1? Fis1 could be particularly relevant, given that it was recently proposed to directly modulate Mfn activity.

To explore the underlying function of hinge 1, the authors could challenge the cells with the four hinge 1 mutants (S378P, A383V, Q386P and C390F) and analyze their morphology. For example, upon oxidizing/reducing conditions (e.g. Shutt T EMBO rep 2011), upon altered GTP/GDP ratios, in hyperfusion conditions, upon mild fragmentation, Can they find some disease-underlying explanation?

Minor suggestions:

In fig. 1, please include a zoom-in of A similarly to C.

Regarding Fig. 3, the best recovery is presented by S378P, the only residue near to, and not on, loop 1. The other mutants only partially recover from cytosol addition. Can the authors speculate on why they are so similar in vivo? Moreover, what is the explanation for S378P impairment of higher oligomerization (Fig. 5), despite stronger trans interaction (Fig. 4)? Further, while S378P shows the lowest amount in the 320kDa complex after addition of Bax (Fig. 5), it also shows the highest in vitro fusion efficiency after addition of cytosolic fraction (Fig. 3). Vice versa, C390F shows the most complex formation of all the mutants, but the lowest fusion activity. Please comment.

Abstract: The last sentence could be more precise (instead of "this") and expand to a broader impact of their findings.

Table S1: Do the disease mutations occur homo- or heterozygous in patients? This information could be helpful to understand the connection between fusion activity and disease.

Text to Fig. S1: maybe rephrase "near-endogenous levels", and rather focus on clonal populations with similar levels between wt and mutant Mfn2.

Text to 3B: 1st when referring to Hoppins 2011, cite also Shutt T EMBO rep 2011. 2nd, regarding "..., indicating that Mfn2, but not Mfn1, is regulated by the cytosolic factor." This is not necessarily the case, it indeed shows that the cytosol requires Mfn2, but does not allow excluding that it also requires Mfn1.

Title of results chapter to Fig. 5: "...self-assembly...". I suggest rather using "oligomerization" as the complexes may also contain Mfn1, consistent with their conclusion from the previous chapter about poor self-interaction in trans of Mfn2.

Title of results chapter to Fig. 6: "Double hinge mutations reveal that HB1 and HB2 work together in mitochondrial fusion". This suggest that HB1 and HB2 have distinct active functions that work together. While undoubtedly showing that the integrity of both HB1- HB2 hinge is important for mitofusin activity, they cannot show whether this includes an active mechanism or is mainly related to structural integrity of the protein. Therefore, this subtitle should be worded differently. In the discussion: "Together, these data suggest that these amino acid substitutions do not significantly alter the structure of Mfn2." Do the authors mean local structure of each domain? In fact, the dynamism in hinge 1 results in different global structures of Mfn2, as they later mention.

Reviewer #3 (Comments to the Authors (Required)):

The manuscript by Samanas et al reports the assembly and fusion abilities of four Charcot-Marie-Tooth Type 2A (CMT2A) related Mfn2 mutants. Mutations in Mfn2 are causally related to the development of CMT2A. The authors designed a serial of assays to illuminate the importance of these four mutants in nucleotide-dependent assembly and fusion in vitro, which is of significance to researchers in mitochondrial dynamic and pathology fields. I would be supportive of publication.

The authors declared that Ser378, Ala383, Gln386, and Cys390 located within or adjacent to hinge1 of Mfn2, which is a starting point. According to figure 1C, the location of these four amino acids was predicted by a predicted structural model based on structure of BDLP (PDB 2W6D). But I don't notice that emphasis on the "prediction" was placed when appeared at the first time in the manuscript. Is this the case?

The trans interaction between Mfn1 and Mfn2 was enhanced nearly twice in the transition mimic state (GDP-BeF3) compared to the negative control (BeF3). I am confused that most Mfn2-FLAG proteins seemingly were occupied by endogenous Mfn1 in a nucleotide independent manner. If that was the case, a very small minority of Mfn2-FLAG would be available for hetro or homo type of trans interaction. To clarify this point, further control co-immunoprecipitation experiment maybe useful and make sense. In detail, isolated mitochondria from MFN2-null MEF cells expressing Mfn2-FLAG and mitochondria from Mfns-null MEF cells are mixed to perform immunoprecipitation experiment in three conditions, namely without ligand, with BeF3 or GDP-BeF3.

These four CMT2A related mutants, supposedly located within or adjacent to the hinge 1 region, restored mitochondrial network in Mfn2-null MEFs in vivo, but not in vitro. The fusion deficiency can be partially rescued by adding cytosolic fraction, which is consistent with Hoppins' result1. Understandably, the fusion deficiency of these four mutants may be caused by diminished nucleotide dependent oligomerization. The deficiency can be rescued by cytosolic factors, like Bax. These works provide insight into the functional affection of domain rearrangement that may result in CMT2A, although further proof need to be offered to confirm the hinge 1 location of Ser378, Ala383, Gln386, and Cys390.

Minor comments:

1. Figure 4B. What's the first line next to "WT"?

2. Was the fusion buffer containing beryllium fluoride or GDP in the mitochondrial mixture removed before adding lysis buffer with 1.5% w/v n-Dodecyl β -D-maltoside (DDM) in the co-

immunoprecipitation experiment? Please indicate this in the method.

1. Hoppins S, et al. The soluble form of Bax regulates mitochondrial fusion via MFN2 homotypic complexes. *Molecular cell* 41, 150-160 (2011).

We would like to thank yourself and the reviewers for your time and careful consideration of our manuscript. We are very pleased that the reviewers acknowledge our findings and we have addressed the issues raised during review as outlined in our point-by-point below. We are excited to submit a revised version of the manuscript with changes in the text highlighted in blue, as well as a clean version.

During revision of our manuscript, a new partial structure of Mfn2 was published (Li et al., Nat. Comm. 2019 PMID 31664033). This atomic resolution structure was obtained for an internally truncated construct of Mfn2, which has the globular GTPase domain and helical bundle 1 (HB1) but lacks the hinge and HB2 (amino acids 401 – 705). In this context, the CMT2A-associated amino acid positions that we have described in this manuscript are all in alpha helix three of HB1. As outlined in our original submission, we modeled the structure of full-length Mfn2 based on homology to BDLP. In this model, the amino acid positions were predicted to be within two distinct helices from HB1 and HB2 and the flexible loop connecting them. The data together suggests that the region is conformationally dynamic. Given this, we would like to revise our language classifying these CMT2A-associated positions as *hinge variants* and thus revise the title of our manuscript to “Defective nucleotide dependent assembly and membrane fusion in Mfn2 CMT2A variants improved by Bax”. Importantly, our data remains the same, even strengthened by the revision, and clearly implicate this region in the GTP-dependent assembly of Mfn2 and its fusion efficiency.

We have added data to Figure 3 to include the stimulation of mitochondrial fusion by purified Bax protein. Although some negative data is included only in our point-by-point response to reviewer comments, we have added the follow supplemental figures:

- Figure S1 Mfn2_{IM} structure with Mfn2 CMT2A-associated variant positions in alpha helix 3.
- Figure S2 Mitochondrial connectivity and fusion as measured by redistribution of GFP in cells.
- Figure S3 Mitochondrial morphology in cells following oxidative stress.
- Figure S4 Immunoprecipitation of Mfn1-eGFP requires Mfn2-FLAG.

Reviewer #1 (Comments to the Authors (Required)):

The authors have studied the role of mutations in the Hinge 1 region of Mfn2 involved in disease on the fusion activity of the protein. They report that these mutations decrease the fusion activity and the oligomerization of the protein in in vitro assays of mitochondria fusion, but mitochondrial shape is not affected in cells. They argue that cytosolic factors contribute to compensate for the fusion defects of the mutants in cells, and indeed show that added Bax is able to increase oligomerization, although this remains as a rather enigmatic observation. Furthermore, the authors show that combination of mutations in both helical bundles of Mfn2 exhibit more severely impaired function. The manuscript is well and clearly written, although the abstract could improve in clarity. The results are in principle of interest for the field, but there are a number of issues that should be addressed before publication.

We thank the reviewer for their time and thoughtful comments on our manuscript. We have revised the abstract to improve clarity.

The authors could explain in more detail which mutations are associated with which diseases. All of the mutant variants characterized in this study are associated with CMT2A. Brief clinical descriptions are in Table 1 in the supplemental material. We have added L710P to this table. For clarity in the main text we have changed "disease-associated" to "CMT2A-associated" variants of Mfn2.

Could the cytosolic factor that masks the lower activity of the Mfn mutants Bax or Drp1? The authors could add the recombinant proteins in the in vitro fusion experiments. This is an excellent suggestion. We have added purified Bax protein to the in vitro fusion assay and we observe stimulation of in vitro fusion activity for each hinge variant (Figure 3C).

Or could it be that the mitochondrial shape is maintained by changes in dynamics lost in isolated mitochondria? To test this later hypothesis the authors could measure mitochondrial dynamics by FRAP or similar. We have assessed mitochondrial fusion and connectivity in cells by monitoring the dispersal of PAGFP from a region of interest into the mitochondrial network over time. This data is now presented in Supplemental Figure 2. We find that in wild type cells, there is a ~2.5 fold increase in the percent of pixels in the mitochondrial network (MitoTracker Red CMXRos) that are also green (mito-PAGFP) 50 minutes following activation of the PAGFP. In Mfn2-null cells expressing a hinge-proximal variant, the fold increase is slightly lower; however, these differences were not statistically significant based on paired t-test analyses.

The authors state: "Together, these data support the conclusion that impaired nucleotide-dependent assembly results in diminished in vitro fusion activity and that cytosolic factors such as Bax compensate for these defects in cells." However they only show correlation, not causality. And for Bax, only a compensation in assembly, not fusion activity, which could easily be tested as mentioned above.

As stated above, we have added purified Bax protein to the in vitro fusion assay and we observe stimulation of in vitro fusion activity for each hinge-proximal variant (Figure 3C).

It is surprising that the authors do not find any mitochondrial defect in the cells expressing the Mfn mutants, given that they are associated with disease. One possibility could be that this is due to the cellular system used, which does not recapitulate the situation in neurons. To test this hypothesis, the authors could check the activity of the Mfn mutants in neurons or neuron-like cells. In addition, they should check if additional mitochondrial functions other than fusion are affected in cells expressing the Mfn mutants, such as metabolism, calcium signalling, mitophagy or apoptosis.

We agree that we cannot use mouse embryonic fibroblasts to model CMT2A. The neurons affected in patients are the longest neurons in the human body and mitochondrial function is likely to be highly specialized in these cells. Our approach was designed to facilitate biochemical analysis of the mitofusin proteins. Unfortunately, we do not currently culture neurons in our lab and given the time constraints of submitting a revised manuscript, we cannot establish this new model system to address these concerns.

However, we have performed a battery of tests to assess mitochondrial and Mfn2-specific function in our existing cells, as outlined below.

(1) We have performed a real time ATP production rate assay using the Seahorse Analyzer (data are shown to the right). Representative Seahorse plots of the oxygen consumption rate (OCR)(Panel A) and the total proton efflux rate (PER)(Panel B) in the indicated cells lines are shown. These data allow the quantification of ATP production from both mitochondria and glycolysis in these cells (Panel C).

Together, these data indicate that there is no significant difference in ATP production in cells expressing the Mfn2 hinge variants compared to wild type.

(2) We measured the total mitochondrial calcium store in controls versus the hinge-proximal variants. Cells plated in black-walled 96-well dishes were loaded with Fura-2AM in HBS with CaCl_2 for 30 minutes at 37°C and 5% CO_2 . Cells were washed with experimental buffer. Using a Gemini XPS dual monochromator fluorescence microplate reader (Molecular Devices), the fluorescence at 340 nm and 380 nm was measured. This represents the basal cytosolic calcium. To measure mitochondrial calcium, cells were treated with CCCP to stimulate release from the mitochondria to the cytosol and the fluorescence was measured again. The increase in fluorescence ratio due to calcium efflux from the mitochondria is plotted below. We observed no significant difference in total mitochondrial calcium in Mfn2-null cells expressing the hinge-proximal variants compared to wild type controls.

(3) To assess mitochondrial fusion under stress conditions, we have quantified mitochondrial hyperfusion in response to diamide, as previously reported (Shutt et al., 2012). We treated cells with diamide and scored mitochondrial morphology in blinded experiments. This data is shown below and presented in Supplemental Figure 3. We do not observe a significant difference in mitochondrial hyperfusion in Mfn2-null cells expressing the hinge-proximal variants compared to wild type controls.

(4) To determine if a fusion defect sensitized the Mfn2-null cells expressing the hinge-proximal variants to cell death, we scored cell viability following treatment with 2 mM etoposide for 16 hours. The data are shown below. We observed no significant difference between Mfn2-null cells expressing Mfn2-WT and the cells expressing the hinge variants. Therefore, we did not detect increased sensitivity to apoptotic cell death.

(5) It was recently reported that lipid droplets accumulated in fibroblasts isolated from Mfn2-associated CMT2A patients, which the authors attributed to mitochondrial associated membrane (MAM) dysfunction (Larrea et al., 2019). As described in Larrea et. al., we stained lipid droplets in our cells with HCS LipidTox Deep Green neutral lipid stain (Invitrogen H34475) according to manufacturer's instructions. Cells were imaged at 37°C with 5% CO₂ on a Nikon Ti-E widefield microscope with a 63X NA 1.4 oil objective (Nikon), a solid state light source (SPECTRA X, Lumencor) and an sCMOS camera (Zyla 5.5 Megapixel). Each cell line was imaged at least two times and the fluorescence intensity was determined using Nikon Elements software. The relative fluorescence intensity compared to wild type controls is shown below. We did not detect a significant difference in the lipid droplet staining in Mfn2-null cells expressing Mfn2-WT and those expressing the hinge-proximal variants. Therefore, we do not find a defect in MAM function in our cells.

Reviewer #2 (Comments to the Authors (Required)):

In the present manuscript, the authors have analyzed the role of CMT2A disease mutants of Mfn2, located at the hinge 1 region, based on a model to the bacterial homologue BDLP. Consistent with previous results from the Dorn and Cohen/Taly labs, among others, they suggest that dynamic conformational changes and oligomerization properties in Mfn2 constitute a critical feature for their functionality. They newly show that cellular compensatory mechanisms allow maintaining mitochondrial reticulation of hinge 1 mutants, despite defects observed when using an in vitro fusion assay. This is concomitant with alterations in mitofusin oligomerization of all hinge 1 mutants analyzed. Thus, the title provided is a totally fair judgment of their results, which are equally globally convincing. However, for a strong conceptual advance, it would be important to provide a mechanistic justification for the "opposing" phenotypes of the CMT2A variants in vivo and in vitro, and resulting disease-implications.

We thank the reviewer for acknowledging our findings and for their helpful comments on our manuscript.

Specific points to be addressed:

Do the four hinge 1 mutants (S378P, A383V, Q386P and C390F) alter fusion in vivo? This should be measured, e.g. with PA-GFP assays.

As stated above, we have assessed mitochondrial fusion and connectivity in cells by monitoring the dispersal of mito-PAGFP from a region of interest into the mitochondrial network over time. This data is now presented in Supplemental Figure 2. We find that in wild type cells, there is a ~2.5 fold increase in the percent of pixels in the mitochondrial network (MitoTracker Red CMXRos) that are also green (mito-paGFP) fifty minutes following activation of PAGFP. In Mfn2-null cells expressing a hinge-proximal variant, the fold increase is slightly lower; however, these differences were not statistically significant as determined by a paired t-test.

In Fig. 3, which is the justification for clear alterations in fusion, as demonstrated by the assay in vitro?

We have previously observed that mitochondrial fusion defects can be masked in cells (Engelhart and Hoppins, 2019).

How do stronger Mfn1-Mfn2 interactions in trans (Fig. 4) (perhaps reflecting longer resident times before hydrolysis) relate to the oligomers observed by BN-page in Fig. 5? In other words, what do these oligomers represent?

Based on the work presented here and in two other papers from our lab (Engelhart and Hoppins, 2019 & Sloat et al., 2019), we believe that the BN-PAGE oligomers are complexes that form in cis, or the same membrane. Several lines of evidence support this conclusion. First, the BN-PAGE complexes are very abundant, whereas the trans complex is not. Second, we only see formation of the trans complex by co-immunoprecipitation in the presence of the transition state mimic (GDP•BeF3). The trans interaction was not detectable in the presence of GMPPNP. Since we see robust assembly in the presence of GMPPNP by BN-PAGE, we believe that the BN-PAGE oligomers are in cis. Third, we cannot detect Mfn2 homotypic trans complex assembly by co-immunoprecipitation but observe robust oligomerization by BN-PAGE. Together, these results indicate to us that BN-PAGE oligomers are homotypic mitofusin protein in cis.

Do the different oligomerization properties of the mutants reflect cis or trans interaction defects?

Since we see no defect in trans complex formation by co-immunoprecipitation, we conclude that the defects are in cis assemblies.

And homo Mfn2 or hetero Mfn1/2 oligomers? This could be addressed by testing the differently tagged variants used for Fig. 4.

Previous work from our lab demonstrates that BN-PAGE measures homotypic complexes as the Mfn1 complexes were not dependent on the presence of Mfn2 and the Mfn2 complexes were not dependent on Mfn1 (Engelhart and Hoppins, 2019 and Sloat et al., 2019).

Consequently, which complex does Bax affect? Cis or trans?

Given that we do not see a defect in the formation of a trans complex by co-immunoprecipitation (Figure 4), we believe that Bax is promoting cis-assembly of Mfn2 as visualized by BN-PAGE (Figure 5).

Before or after hydrolysis?

The effect of Bax on Mfn2 assembly was tested in the presence of a non-hydrolysable nucleotide analog (GMPPNP); therefore, we believe the assembly represents a state that occurs before nucleotide hydrolysis.

Measurements of GTP binding and hydrolysis rates could help explaining these mutants.

Accurate rates of GTP binding and hydrolysis are challenging in the absence of biochemically purified protein. Unpublished data from our lab indicates that we cannot detect robust GTPase activity with recombinant purified Mfn2. Consistent with this, previous reports indicate that Mfn2 is a poor enzyme and a recent report of a minimal catalytic domain supports this conclusion. Furthermore, data from our labs and others indicate that catalytic dead variants of Mfn2 can support fusion in the presence of Mfn1. For these reasons, we did not measure GTP binding or hydrolysis here.

Further, in Fig. 4, the effect observed by Bax addition for oligomerization is rather mild. A convincing role of Bax in these Mfn2 mutants should be demonstrated in cells, perhaps with experiments reminiscent of their own Mol Cell paper. E.g., how does soluble/mito-inserted/apoptosis-incompetent Bax affect the mitochondrial morphology of these 4 Mfn2 mutants?

As stated above, we have added purified Bax protein to the in vitro fusion assay and we observe stimulation of in vitro fusion activity for each hinge variant. These data are presented in Figure 3C.

In Fig. S1, please include a shorter exposure of both blots.

The images have been exchanged so that a lower exposure is shown.

In Fig. 3, panels A and B could be presented with the same scaling, and the average values indicated.

The graph in Figure 3A represents the relative fusion of the mitochondria with an Mfn2-CMT2A variant compared to wild type mitochondrial fusion reactions performed in parallel. For clarity, we have changed the Y-axis to “percent wild type fusion”. In contrast, Figure 3B is fold stimulation of mitochondrial fusion, which is a relative expression of in vitro fusion of the same mitochondria in the presence/absence of cytosol or Bax. Given this, we do not believe it is appropriate to express these as the same scale as the graph in 3A.

In Fig. 4, a Mfn1 and Mfn2 antibody specificity should be included. Plus, a negative control with cells w/o Mfn2-Flag should be performed, to subtract background binding of Mfn1-eGFP to the beads.

We have adjusted the labels on Figure 4 to clearly identify each Mfn1 and Mfn2 species on the western blot. We have also performed the trans co-immunoprecipitation using wild type mitochondria (lacking Mfn2-FLAG) and Mfn1-eGFP mitochondria. In the absence of Mfn2-FLAG, we do not observe any Mfn1-eGFP in the eluate. These data are shown below and are now included in the manuscript as Supplemental Figure 4. Notably, the results of the immunoprecipitation are variable in biological replicates and the Western blot presented in Figure 4 is a representative image of a day when we saw some eGFP in the negative control, which was observed on multiple occasions.

In Fig. 5A, a negative control w/o Mfn2-Flag should be added. Also, a short exposure of the blots should be included.

The BN-PAGE gels are technically challenging. We have included an image here with a shorter exposure for all samples in Figure 5A. We have chosen to leave the original image in the manuscript as we believe it is slightly easier to see all of the relevant species than the shorter exposure. The wild type mitochondria western blot with FLAG antibody is also shown here.

To further broaden the conceptual advance, the following questions might be considered:

Do their mutants alter fusion/fission balance? Is fission inhibited? Do the mutant cells respond/interact differently to Drp1, Drp1 receptors, or Fis1? Fis1 could be particularly relevant, given that it was recently proposed to directly modulate Mfn activity.

In time lapse live cell imaging experiments, we found that mitochondrial division events were frequent in Mfn2-null MEFs expressing a hinge-proximal variant. Cells were stained with Mitotracker Red CMXRos and subsequently visualized with fluorescence microscopy at 37°C and with 5% CO₂. A Z-series with a step size of 3 microns was collected every 15 seconds for 10 minutes. Three representative examples (labeled A, B, C) of mitochondrial division in Mfn2-null cells expressing the indicated variant are shown here. The position of the division event is highlighted with a yellow arrow.

We have tested for a physical interaction between Mfn2-FLAG and Fis1 in our cells.

Mitochondria isolated from Mfn2-null cells expressing either Mfn2-WT, Mfn2-S378P, Mfn2-A383V, Mfn2-Q386P or Mfn2-C390F or mitochondria isolated from wild type cells (WT) were

solubilized on ice in lysis buffer (50 mM HEPES KOH pH 7.4, 50 mM KCl, 5 mM MgCl₂, 1 mM GTP + 1.5% digitonin and protease inhibitor cocktail). Following centrifugation to remove insoluble fraction, the supernatant was incubated with 50 μ L MACS anti-DYKDDDDK Microbeads (Miltenyi Biotec) for 30 minutes on ice. The sample was applied to a μ MACS column placed in the magnetic field using a μ MACS separator and washed once with lysis buffer, twice with wash buffer I (50 mM HEPES KOH pH 7.4, 50 mM KCl, 5 mM MgCl₂ + 0.1% digitonin) and once with wash buffer II (50 mM HEPES KOH pH 7.4, 50 mM KCl, 5 mM MgCl₂). One column volume of SDS-PAGE loading buffer was added to the column and incubated for 10 minutes before eluting in 35 μ L SDS-PAGE loading buffer.

As a positive control for co-immunoprecipitation, we blotted for Mfn1. As negative controls, we blotted for Hsp60 and VDAC. In contrast to Mfn1, which we can detect in the elution fraction, we did not detect Fis1 interacting with Mfn2-WT or any of the hinge-proximal variants.

To explore the underlying function of hinge 1, the authors could challenge the cells with the four hinge 1 mutants (S378P, A383V, Q386P and C390F) and analyze their morphology. For example, upon oxidizing/reducing conditions (e.g. Shutt T EMBO rep 2011), upon altered GTP/GDP ratios, in hyperfusion conditions, upon mild fragmentation, Can they find some disease-underlying explanation?

As stated above (Page 4) we have assessed mitochondrial fusion under stress conditions, we have quantified mitochondrial hyperfusion in response to diamide, as previously reported (Shutt et al., 2012). We treated cells with diamide and scored mitochondrial morphology in blinded experiments. This data is shown above and presented in Supplemental Figure 3. We do not observe a significant difference in mitochondrial hyperfusion in Mfn2-null cells expressing the hinge-proximal variants compared to wild type controls.

Minor suggestions:

In fig. 1, please include a zoom-in of A similarly to C.

Regarding Fig. 3, the best recovery is presented by S378P, the only residue near to, and not on, loop 1. The other mutants only partially recover from cytosol addition. Can the authors speculate on why they are so similar in vivo?

We speculate that our in vitro mitochondrial fusion assay is more sensitive to small changes in protein function and that other factors missing from the assay (tension provided by tethering to

microtubules) are also contributing to the difference. We also have preliminary, unpublished data that there are cytosolic protein(s) in addition to Bax that contribute to mitochondrial fusion in our assay.

Moreover, what is the explanation for S378P impairment of higher oligomerization (Fig. 5), despite stronger trans interaction (Fig. 4)?

As outlined above, we believe that the BN-PAGE and trans co-immunoprecipitation are measuring two different assembly pathways for mitofusin.

Further, while S378P shows the lowest amount in the 320kDa complex after addition of Bax (Fig. 5), it also shows the highest in vitro fusion efficiency after addition of cytosolic fraction (Fig. 3). Vice versa, C390F shows the most complex formation of all the mutants, but the lowest fusion activity. Please comment.

As shown in Figure 3D, the new data quantifying fold stimulation of mitochondrial fusion in the presence of purified Bax protein is consistent with the BN-PAGE data. Mfn2-S378P shows the lowest degree of stimulation by purified Bax proteins compared to the other hinge mutants.

Abstract: The last sentence could be more precise (instead of "this") and expand to a broader impact of their findings.

We have adjusted the text to read "Our data are consistent with a model where the hinge region contributes to conformational changes that are important for assembly and that cytosolic factors, such as Bax, are compensating for fusion defects in these CMT2A variants."

Table S1: Do the disease mutations occur homo- or heterozygous in patients? This information could be helpful to understand the connection between fusion activity and disease.

In all cases, the patients are heterozygous for the mutation. This has been added to the text to clarify.

Text to Fig. S1: maybe rephrase "near-endogenous levels", and rather focus on clonal populations with similar levels between wt and mutant Mfn2.

We have adjusted the text accordingly.

Text to 3B: 1st when referring to Hoppins 2011, cite also Shutt T EMBO rep 2011. 2nd, regarding "..., indicating that Mfn2, but not Mfn1, is regulated by the cytosolic factor." This is not necessarily the case, it indeed shows that the cytosol requires Mfn2, but does not allow excluding that it also requires Mfn1.

We have adjusted the text accordingly.

Title of results chapter to Fig. 5: "...self-assembly...". I suggest rather using "oligomerization" as the complexes may also contain Mfn1, consistent with their conclusion from the previous chapter about poor self-interaction in trans of Mfn2.

We have adjusted the text accordingly.

Title of results chapter to Fig. 6: "Double hinge mutations reveal that HB1 and HB2 work together in mitochondrial fusion". This suggest that HB1 and HB2 have distinct active functions that work together. While undoubtedly showing that the integrity of both HB1- HB2 hinge is

important for mitofusin activity, they cannot show whether this includes an active mechanism or is mainly related to structural integrity of the protein. Therefore, this subtitle should be worded differently.

We have adjusted the text to read "Double hinge mutations reveal that HB1 and HB2 contribute independently to mitochondrial fusion".

In the discussion: "Together, these data suggest that these amino acid substitutions do not significantly alter the structure of Mfn2." Do the authors mean local structure of each domain? In fact, the dynamism in hinge 1 results in different global structures of Mfn2, as they later mention.

We meant to convey that the protein is generally well-folded and stable for each variant. We have adjusted the text to read "these data suggest that these amino acid substitutions do not alter protein folding or stability."

Reviewer #3 (Comments to the Authors (Required)):

The manuscript by Samanas et al reports the assembly and fusion abilities of four Charcot-Marie-Tooth Type 2A (CMT2A) related Mfn2 mutants. Mutations in Mfn2 are causally related to the development of CMT2A. The authors designed a serial of assays to illuminate the importance of these four mutants in nucleotide-dependent assembly and fusion in vitro, which is of significance to researchers in mitochondrial dynamic and pathology fields. I would be supportive of publication.

The authors declared that Ser378, Ala383, Gln386, and Cys390 located within or adjacent to hinge1 of Mfn2, which is a starting point. According to figure 1C, the location of these four amino acids was predicted by a predicted structural model based on structure of BDLP (PDB 2W6D). But I don't notice that emphasis on the "prediction" was placed when appeared at the first time in the manuscript. Is this the case?

We thank the reviewer for their helpful comments on our manuscript. We have modified our language regarding the position of these amino acids and state that we cannot know where each is in full-length Mfn2.

The trans interaction between Mfn1 and Mfn2 was enhanced nearly twice in the transition mimic state (GDP·BeF3) compared to the negative control (BeF3). I am confused that most Mfn2-FLAG proteins seemingly were occupied by endogenous Mfn1 in a nucleotide independent manner. If that was the case, a very small minority of Mfn2-FLAG would be available for hetro or homo type of trans interaction. To clarify this point, further control co-immunoprecipitation experiment maybe useful and make sense. In detail, isolated mitochondria from MFN2-null MEF cells expressing Mfn2-FLAG and mitochondria from Mfns-null MEF cells are mixed to perform immunoprecipitation experiment in three conditions, namely without ligand, with BeF3 or GDP·BeF3.

As stated above, we have also performed the trans co-immunoprecipitation using wild type mitochondria (lacking Mfn2-FLAG) and Mfn1-eGFP mitochondria. In the absence of Mfn2-

FLAG, we do not observe any Mfn1-eGFP in the eluate. These data are shown above (page 8) and are now included in the manuscript as Supplemental Figure 4.

These four CMT2A related mutants, supposedly located within or adjacent to the hinge 1 region, restored mitochondrial network in Mfn2-null MEFs in vivo, but not in vitro. The fusion deficiency can be partially rescued by adding cytosolic fraction, which is consistent with Hoppins' result¹. Understandably, the fusion deficiency of these four mutants may be caused by diminished nucleotide dependent oligomerization. The deficiency can be rescued by cytosolic factors, like Bax. These works provide insight into the functional affection of domain rearrangement that may result in CMT2A, although further proof need to be offered to confirm the hinge 1 location of Ser378, Ala383, Gln386, and Cys390.

We thank the reviewer for acknowledging our findings.

Minor comments:

1. Figure 4B. What's the first line next to "WT"?

In the western blot analysis of the trans co-immunoprecipitation, Mfn2-S378P is loaded next to wild type.

2. Was the fusion buffer containing beryllium fluoride or GDP in the mitochondrial mixture removed before adding lysis buffer with 1.5% w/v n-Dodecyl β -D-maltoside (DDM) in the co-immunoprecipitation experiment? Please indicate this in the method.

We have adjusted the methods to include this information. The lysis buffer does not contain the transition state mimic, but it was not removed from mitochondria prior to lysis.

1. Hoppins S, et al. The soluble form of Bax regulates mitochondrial fusion via MFN2 homotypic complexes. Molecular cell 41, 150-160 (2011).

March 24, 2020

RE: Life Science Alliance Manuscript #LSA-2019-00527R

Dr. Suzanne Hoppins
University of Washington
1959 NE Pacific St Health Sciences Building J383
Seattle, WA 98195

Dear Dr. Hoppins,

Thank you for submitting your revised manuscript entitled "Defective nucleotide dependent assembly and membrane fusion in Mfn2 CMT2A variants improved by Bax". As you will see, the reviewers appreciate the introduced changes. Reviewer #2 still suggests to include in cellulo assays with Bax and Mfn2 mutants to analyze effects on mitochondrial morphology. However, since we pre-discussed the revision, including such further analysis is not mandatory for acceptance here. I would thus be happy to publish your paper in Life Science Alliance, and I would like to invite you to upload final files / move all files to the next version, and to fill in our electronic license to publish form.

A. FINAL FILES:

B. MANUSCRIPT ORGANIZATION AND FORMATTING:

Sincerely,

Reviewer #1 (Comments to the Authors (Required)):

The authors have addressed the reviewers concerns adequately and the manuscript is acceptable for publication.

Reviewer #2 (Comments to the Authors (Required)):

In this revised version, the authors addressed nearly all concerns raised and I am favorable of publication. A big number of hypotheses to further understand these CMT2A-mutant variants have been addressed. I would suggest including these data as supplements in the manuscript. The only important point still unaddressed is the consequence of Bax absence for mitochondrial morphology, in cells expressing the Mfn2 mutants. I would highly recommend still including it, especially given their strengthen conclusions regarding the importance of Bax in vitro.

March 25, 2020

RE: Life Science Alliance Manuscript #LSA-2019-00527RR

Dr. Suzanne Hoppins
University of Washington
1959 NE Pacific St Health Sciences Building J383
Seattle, WA 98195

Dear Dr. Hoppins,

Thank you for submitting your Research Article entitled "Defective nucleotide dependent assembly and membrane fusion in Mfn2 CMT2A variants improved by Bax". It is a pleasure to let you know that your manuscript is now accepted for publication in Life Science Alliance. Congratulations on this interesting work.

DISTRIBUTION OF MATERIALS:

Again, congratulations on a very nice paper. I hope you found the review process to be constructive and are pleased with how the manuscript was handled editorially. We look forward to future exciting submissions from your lab.

Sincerely,
